# Near-Infrared Metabolomic Fingerprinting Study of Lichen Thalli and Phycobionts in Culture: Aquaphotomics of *Trebouxia lynnae* Dehydration

**DOI:** 10.3390/microorganisms10122444

**Published:** 2022-12-10

**Authors:** Irene Bruñas Gómez, Monica Casale, Eva Barreno, Myriam Catalá

**Affiliations:** 1Department of Biology and Geology, School of Experimental Science & Technology, Rey Juan Carlos University, Av. Tulipán s/n, 28933 Madrid, Spain; 2DIFAR Department of Pharmacy, University of Genova, 16148 Genova, Italy; 3Instituto Cavanilles de Biodiversidad y Biología Evolutiva (ICBiBE), Botánica, Universitat de València, C/Dr. Moliner, 50, 46100 Valencia, Spain

**Keywords:** anhydrobiosis, desiccation, phycobiont, *Ramalina farinacea*, *Lobarina scrobiculata*, *Trebouxia lynnae*, *Trebouxia jamesii*, aquaphotomics, metabolomic profile

## Abstract

Near-infrared spectroscopy (NIRS) is an accurate, fast and safe technique whose full potential remains to be exploited. Lichens are a paradigm of symbiotic association, with extraordinary properties, such as abiotic stress tolerance and adaptation to anhydrobiosis, but subjacent mechanisms await elucidation. Our aim is characterizing the metabolomic NIRS fingerprints of *Ramalina farinacea* and *Lobarina scrobiculata* thalli, and of the cultured phycobionts *Trebouxia lynnae* and *Trebouxia jamesii*. Thalli collected in an air-dry state and fresh cultivated phycobionts were directly used for spectra acquisition in reflectance mode. Thalli water peaks were associated to the solvation shell (1354 nm) and sugar–water interactions (1438 nm). While northern–southern orientation related with two hydrogen bonded (S_2_) water, the site was related to one hydrogen bonded (S_1_). Water, lipids (saturated and unsaturated), and polyols/glucides contributed to the profiles of lichen thalli and microalgae. *R. farinacea*, with higher desiccation tolerance, shows higher S_2_ water than *L. scrobiculata*. In contrast, fresh phycobionts are dominated by free water. Whereas *T. jamesii* shows higher solvation water content, *T. lynnae* possesses more unsaturated lipids. Aquaphotomics demonstrates the involvement of strongly hydrogen bonded water conformations, polyols/glucides, and unsaturated/saturated fatty acids in the dehydration process, and supports a “rubbery” state allowing enzymatic activity during anhydrobiosis.

## 1. Introduction

Metabolism refers to all chemical reactions that occur inside the cells of a living being [1]. Although current metabolomics techniques usually focus on the study of the smallest metabolites, the concept of “metabolite” applies to any product of metabolism, regardless of its molecular size [2]. The objective of metabolomics is to determine the profile of the metabolites of a specimen in order to infer the metabolic pathways active in each physiological situation. It has been applied to various fields of clinical research, such as biomarker discovery or toxicology, but it can also be useful for taxa fingerprint identification [3].

Typically, chromatographic methods, such as liquid chromatography mass spectrometry (LCMS) [4] or gas chromatography mass spectrometry (GCMS) [5], are used to evaluate the metabolites in the samples. These methods are well known and conventional, as they have several advantages, such as their results being very accurate. Among their disadvantages, it should be noted that these methods require long processing and acquisition times, and, in addition, they use solvents that are toxic to the operator or the environment. For this reason, alternative green methods have been sought that can predict the compounds in the sample, such as near-infrared spectroscopy (NIRS). One of the most important advantages of this method is that the sample can be measured non-destructively. Other advantages of NIRS are that it is fast, accurate, has a low cost of analysis, and is environmentally friendly because it does not use or generate chemical waste. It is also possible to measure many samples in a very short period [6,7].

Fourier Transform Near-Infrared Spectrophotometry (FT-NIRS) is a classical technique that has been in use since the invention of spectroscopy. However, its use in biodiversity research is recent, and its application to metabolomic phenotype characterization under abiotic stress is much more novel [8].

FT-NIRS combines near-infrared spectroscopy (NIRS) with different mathematical treatments [9]. It is based on the interaction that occurs between an incident beam of light and the surface of a material [10]. The Lambert–Beer law, which states that the amount of light absorbed depends on the concentration of the substance or body being measured, is always observed. The wavelength range used for the energy pulses ranges from 780 nm to 2500 nm, causing the vibration of the molecules, which give rise to the appearance of absorbance bands. The most frequent bands in this range correspond to overtones and combinations of the fundamental mid-infrared (IR) molecular vibration bands. Overtones are vibrational transitions from the ground state to higher excited states, and band combinations result from the coupling of two bands. These vibrations are related to the different molecular bonds present in the sample: S-H, C=O, O-H, C-H, C-H and N-H [11].

The sample to be measured in NIRS can be liquid, gaseous, or solid. For each wavelength value, the radiation is absorbed by the bonds. At the same time, light with a different wavelength is either transmitted or reflected. All the information obtained from the instrument is processed using a computer and a spectrum is obtained which is like its global molecular fingerprint because it is unique for each sample [12]. It is a very valuable tool that gives an immediate picture of the relative content of all the metabolomic groups of the sample at the same time [13]. This fingerprint helps to understand the different properties of the sample, whether physical, biological, or chemical, and allows discrimination from the other samples because it is unique [14].

Lichens are a clear example of a symbiotic association, consisting of the interactions of very complex systems that originate from the coexistence of, at least, two main organisms: a photosynthetic one, that can be a microalga or a cyanobacterium, and a structural one, that is a fungus [15]. The photosynthetic part is the main producer because it supplies the fungus with the carbohydrates obtained by photosynthesis [16], while the fungus gives the microalgae shelter, water and minerals. The most sensitive part of the lichen is the photobiont, as has been shown in various studies [17,18,19]. Thanks to this symbiotic association, lichens have been able to colonize desert areas due to their desiccation tolerance (DT) [20].

Lichens are poikilohydric; that is, they do not have mechanisms to regulate their water content, and for that reason, they depend directly on the water in the environment. When the thallus is hydrated, the photobiont manufactures photosynthetic carbohydrates that are used for its own growth, but also as food for the fungus. On the other hand, when the thallus completely loses water, it enters a state called anhydrobiosis. Lichens can withstand infinite cycles of desiccation/rehydration (D/R). The only important thing for this to happen is that they are hydrated long enough to balance the energy cost produced by the respiration of the mycobiont and produce carbohydrates for it to grow. This behavior is lethal for most organisms, but lichens have certain biochemical, physiological, and molecular mechanisms that help them to avoid cell damage during desiccation.

In the works of Barreno and co-workers, it has been shown that both the morphology and the photosynthetic function of lichens depend on their water content and their ability to retain water [21,22]. Lichen tolerance to contaminants is also affected by the availability of water in the environment [23] since there is a dependence between photosynthetic activity and the level of turgor due to the water stress [24]. Lichens have different ways of obtaining water in addition to rain, such as dew or the simple humidity present in the environment, and in this way they can prolong their rehydration time. In addition, they are capable of withstanding very extreme temperature ranges, both high and low, as well as ultraviolet radiation, dehydration, and sunstroke. For these reasons, we can find lichens in environments as extreme as mountains or deserts [25]. Lichens are known to occupy 7% of the earth’s surface and are found from polar regions to the tropics [26]. Two well-known lichens are: *Ramalina farinacea* (L.) Ach., a Mediterranean-Atlantic to southern boreal lichen found on humid sites, and *Lobarina scrobiculata*- (Scop.) Cromb, a suboceanic species of temperate climate, found on old deciduous trees and on mossy rocks in humid open forests [27].

Studies carried out with the phycobionts (green algae photobionts) *Trebouxia lynnae* Barreno (formerly known as *Trebouxia* sp. TR9) and *Trebouxia jamesii* (Hildreth & Ahmadijian) Gärtner, isolated from the lichen *Ramalina farinacea* [28], demonstrate that *T. lynnae* is highly resistant to salt and osmotic stress, as well as water loss, so much so that no changes in its morphology, coloration, and growth are apparent [29]. Previous studies have shown that *T. lynnae* loses water faster when relative humidity is low, suggesting that the behavior of this phycobiont is different from that of *T. jamesii* [30]. Casano and co-workers have analyzed that aero-terrestrial microalgae show a preference for habitats with less water due to their ability to trigger their antioxidant system and their tolerance to desiccation [31].

Aquaphotomics means “all about water–light interaction”. This is a particularly new branch of spectrophotometry and, despite the number of works published on different biological models, its full potential is yet to be exploited. Water absorbs across the entire electromagnetic spectrum, and the water–light interaction can be used to study the changes caused by the energy of various frequencies on the molecular system of water. It is useful in the near-infrared range because light is neither completely reflected (as in the visible) nor absorbed (as in the infrared). It is very informative for bioaqueous systems. In the NIR range, water has a specific spectral pattern that changes with perturbations, and the main goal of aquaphotomics is to build a database of the absorbance bands of water, called aquaphotome, and thus to be able to use them as biomarkers to determine how the system is functioning [32]. Most studies that have been conducted in the NIR focus on the first O-H overtone (1300–1600 nm) because this is where most water bands are identified [32,33,34]. Despite the advantages of NIRS, the spectra obtained are very complex to interpret and analyze. For this reason, chemometrics is used to resolve such complex results [35]. The most common chemometric method is principal component analysis (PCA). This method is a dimensionality reduction technique that allows for the visualization of the variations present in the dataset and the discovery of hidden trends and patterns, as well as differences among the measured samples [36].

Therefore, the hypotheses of this work are: (1) The comparison of the NIRS molecular fingerprint of different species of thalli provides information on their metabolomic phenotype. (2) Changes in the NIRS fingerprint during the dehydration process of the phycobiont *T. lynnae* and its aquaphotomics will give insights into the associated metabolomic changes, as well as its water molecular structure management.

The general objective of this work is to evaluate by NIRS the metabolomic differences that exist between two species of lichens, and between two of the phycobionts of the lichen *Ramalina farinacea*, as well as the molecular structure of water during *Trebouxia lynnae* dehydration. For this, the specific objectives are: (1) obtaining the NIRS metabolomic fingerprint of each organism, (2) obtaining the metabolomic fingerprint of the phycobionts *Trebouxia lynnae* and *Trebouxia jamesii* in culture, and (3) studying the water molecular organization in *T. lynnae* during the dehydration process, using aquaphotomics.

## 2. Materials and Methods

### 2.1. Biological Material and Culture of the Phycobionts

Thalli of *Ramalina farinacea* and *Lobarina scrobiculata* were collected in an air-dry state in El Escorial (Madrid, Spain) at Ermita de la Virgen de Gracia (40°34′23″ N, 4°9′15″ W, 969 m a.s.l.) in June 2021. Supplementary thalli of *R. farinacea* were collected from a second site of the same population at Silla de Felipe II (40°34′6″ N, 4°9′9″ W, 1057 m a.s.l.). For this species, thalli orientation (north or south) on the tree trunk was noted for both sites. Once they arrived at the laboratory, they were left for 24 h on a bench at a temperature of 21–25 °C and a relative humidity of 30–40%.

Two isolated phycobionts in semi-solid culture were used:-*Trebouxia jamesii TR1*, isolated from the lichen *Ramalina farinacea* from a population of Sierra El Toro (Castellón, Spain; 39°54′16″ N, 0°48′22″ W), previously described in Gasulla et al. [17].-*Trebouxia* sp. *TR9*, isolated from the lichen *Ramalina farinacea* from a population of Sierra El Toro (Castellón, Spain; 39°54′16″ N, 0°48′22″ W), previously described in Gasulla et al., [17]. It has recently been renamed as *Trebouxia lynnae* [37].

Cultures of both species were kindly gifted by Dr. Gasulla and Prof. Barreno. For the liquid medium, Bold’s Basal Medium (3 NBBM) was prepared, as indicated by Ahmadjian in 1960 [38]. Stock cultures of the algae were maintained in 10 mL tubes with 3 NBBM plus 10 g casein and 20 g glucose per liter. The semi-solid medium was obtained by adding 15 g of European bacteriological agar to the liquid medium. Culture media were then autoclaved for 20 min at 121 °C. Cultures on the semisolid medium were grown in 5.2 or 8.5 cm diameter plastic Petri in the culture chamber for 21 days at 19 °C and 12 h of white light with an irradiance of 25 μmol m^−2^s^−1^.

### 2.2. Dehydration of Phycobionts

Approximately 5 g of microalgae were weighed in pre-weighed excavated glass slides. The algae were placed in a plastic box containing silica gel for dehydration. This box was returned to the chamber at a controlled temperature and covered with aluminum foil to avoid light [39,40]. Samples were weighed every hour until a total time of 4 h was reached, which is when the weight stabilized.

### 2.3. Acquisition of NIR spectra

A piece of approximately 2 cm of thallus was placed directly on the NIRS reflectance detector window (Figure 1B) (NIRA reflectance accessory). For the blank, a measurement was made in air. A reflector was placed on top of the thallus to improve the signal. Phycobionts were manipulated on excavated glass slides. In this case, the blank was made with a clean slide.

The samples were measured with a PerkinElmer (Beaconfield, U.K.) Spectrum 100 N Series Fourier Transform Spectrophotometer (Figure 1A). The absorbance scan range was from 1000 nm to 2500 nm. The spectrum obtained is the average of the 32 scans made by the apparatus. The spectral resolution was 16 cm^−1^ (0.5 nm).

### 2.4. Data Analysis

For molecular fingerprint profile characterization, standard normal variate transformation (SNV) [41] was performed to reduce the effects of light scattering phenomena before averaging. Then a smoothing (Savitzky–Golay, Order 3, Points 31) and a 2nd derivative transformation (Savitzky–Golay, Order 3, Points 15) was applied to correct the baseline and make the most relevant peaks of the spectra clearly visible. Peaks were related to different molecular bonds with the help of NIRS theoretical tables [42] and pertinent literature. For the subtraction of the spectra, we also used the mean absorbance of SNV pretreated spectra of either thalli or microalgal cultures.

PCA was performed as a multivariate display method on the NIR spectra of *Ramalina farinacea* thalli or *Trebouxia lynnae* in order to extract the useful information embodied within the data and to visualize the data structure, after SNV and column centering as data pretreatments.

### 2.5. Aquaphotomics

Aquaphotomics analyses were performed on the spectra pretreated with extended multiplicative dispersion correction (EMSC) [43] in groups according to the time of dehydration. With the aid of the literature and our own results presented here, six peaks corresponding to different molecular species of water were identified: S_r_ (1346 nm) clustering of protonated water molecules; S_0_ (1406 nm) free water; S_1_ (1443 nm) water molecules with hydrogen one bond; S_2_ (1464 nm) water molecules with two hydrogen bonds; S_3_ (1490 nm) water molecules with three hydrogen bonds; and S_4_ (1652 nm) water molecules with four hydrogen bonds [44,45]. Data analyses focused on these absorbance bands, corresponding to the different water species in our spectra.

A repeated measures analysis of variance (ANOVA) was applied to test whether the observed differences were significant or not. To find out how much each water species contributes, relative absorbance was calculated for each dehydration time as in the article by Kuroki et al. 2019 [46] with the Formula (1):(1)$$A_{rel,S}=\frac{A_{S}}{\sum_{i}A_{S_{i}}},$$
where $A_{rel,S}$ is the relative absorbance of the water species by percentage, $A_{S}$ is the mean absorbance value for each water species (arithmetic mean), and $\sum_{i}A_{S_{i}}$ is the sum of the mean absorbances of all water species S at that time.

### 2.6. Statistical Analysis

The total number of replicates for thalli measured in NIRS is: *Ramalina farinacea* N = 40, and *Lobarina scrobiculata* N = 96. The number of replicates for *Trebouxia lynnae* and *Trebouxia jamesii* phycobionts was N = 21 for each. In the case of dehydrated *Trebouxia lynnae* phycobionts, the number of replicates was also N = 21.

The processing of the spectra and data was performed with The Unscrambler X 10.4 software and Microsoft Office Excel 2021.

## 3. Results

### 3.1. Lichen Thalli Global Molecular Fingerprint by NIRS

#### 3.1.1. NIRS Spectrum of *Ramalina farinacea*

Figure 2A shows the mean spectrum of the lichen *Ramalina farinacea* together with the replicates after SNV (N = 40). The second derivative applied to the mean spectrum allows us to reveal peaks that are not visible in the untransformed spectrum (Figure 2B). In this way, each peak can be assigned to a metabolomic group by using theoretical assignment tables and literature. Although the thalli were in a physiological air-dry state, two important peaks related to water are observed at 1435, attributed to water with one hydrogen bond (S_1_) [47] and 1920 nm [48], as well as a peak at 1359 nm related with the water solvation shell [49]. Peaks related to lipids (1204 and 1724, 2315) [50,51] and polyols, including glucides (2270 nm), also appear [52].

PCA was performed as a multivariate display method on the NIR spectra of *Ramalina farinacea* thalli, after SNV and column centering as data pretreatment, in order to extract the useful information embodied within the data and to visualize the data structure regarding the orientation on the tree trunk (northern or southern) and site of collection. To verify the repeatability of the measurements and to evaluate the analytical variability, the PCA was performed considering all 40 replicates.

Figure 3A shows the PCA score plot in the space of the first two principal components (PCs), explaining 91% of the total variance without the sixth replica of class 3 (A#6 of Class SillaN), which is to be considered an outlier given the distant position from the other replicas of Silla N. Not all the classes are clearly separated, but samples from Silla North form a cluster rather well separated from the other ones along PC1. According to this observation, PC1 seems to explain information related to the orientation of the samples. Looking at the loading profile on PC1 in Figure 3B, it is possible to see two bands around 1460 nm and 1929 nm having higher loadings on this PC. Both bands are related with water molecules [48], and the first one is specifically related with water with two hydrogen bonds (S_2_) [47]. Three negative bands are also visible around 2141 nm, related to unsaturated lipids, [50] and 2394 nm, which could be related to polyols, including sugars [53].

Finally, Figure 4 shows the PCA score plot in the space of PC2 versus PC3. Samples are represented using the same colors as previously, according to their site and orientation, and it can be seen that samples partially differ according to this information. In the score plot of Figure 3, it is possible to recognize a cluster for the Ermita North samples, all of which are at positive values on PC3, that seems to explain the information related to the site; in fact, most Silla samples are at negative values of PC3, whereas Ermita samples preferentially appear at positive values. In addition, in this case, looking at the loading profile on PC3 in Figure 4, it is possible to see the bands explaining the information mainly related to the specific site. In this case, bands at 1440, related with S_1_ water (one hydrogen bonded molecules) [47], as well as hydronium involved in sugar–water interaction [49], and 2127 nm show positive loadings. Three bands present remarkable negative loadings at 1638, 1731 (related to lipids [51]) and 2261 nm (related to glucides [54]).

#### 3.1.2. NIRS Spectrum of *Lobarina scrobiculata*

Figure 5A shows the mean spectrum of the lichen *L. scrobiculata* in the physiological air-dry state. The gray spectra stand for the replicates that were made and the black spectrum is the mean after SNV transformation. The main peaks are indicated as revealed by the second derivative (Figure 5B). As in the case of *R. farinacea*, although the thalli were in a physiological dry state, two peaks linked to water are observed at 1354 nm (water solvation shell [49]), 1438 (S_1_ water [47], hydronium involved in sugar–water interaction [49]) and 1921 nm [48]. Peaks related to lipids (1210, 1696, 1728) [51], alcohol aliphatic chains (2343 nm) [52,53], proteins (2058 nm) [55], and polyols including glucides (2269 nm) also appear [54].

#### 3.1.3. Subtraction Spectrum of *Ramalina farinacea* and *Lobarina scrobiculata*

The subtraction spectrum was performed by subtracting the absorbance values of the mean spectrum of *L. scrobiculata* from the absorbances of the mean spectrum of *R. farinacea* (Figure 6). Areas above zero indicate that *R. farinacea* has a greater contribution of compounds that absorb in these wavelengths. It appears to have a higher contribution of water molecules and some types of fatty acids. Areas below zero indicate that *R. farinacea* has a lower presence of compounds that absorb at these wavelengths. Thus, it seems that *R. farinacea* has a higher contribution of S_2_ water (bands around 1460 nm and 1920 nm) [48,49] and proteins (band around 2062 nm) [55], and lower contribution from some types of polyols, including glucides (band from 2222 to 2325 nm) [52] and lipids (band from 2300 to 2370 nm) [50].

### 3.2. Spectra of Ramalina farinacea Phycobionts

Since the metabolomic profile of the lichen Ramalina farinacea has been obtained, we are going to study two of its main phycobionts to compare with the thallus from which they originate.

#### 3.2.1. NIRS Spectrum of *Trebouxia lynnae*

Figure 7A shows the mean spectrum of the alga *T. lynnae*. As previously seen, the gray spectra are the replicates after SNV (N = 21) and the black line is the mean. In Figure 7B, the second derivative has been applied to the mean spectrum. Three peaks related to free water are observed at 1151, 1409 and 1900 nm [48]. A new peak at 1151 nm that was not detected in lichen thalli appears due to the high-water content of fresh algae. In addition, peaks attributed to H_2_O asymmetric vibration at 1345 nm and S_2_ water at 1466 nm are clearly visible [47,49]. Peaks related to saturated (1202 nm) and unsaturated fatty acids (2123 nm) [50], and polyols/glucides (2278 nm) [52,54] appear. Other peaks related with aliphatic chains are visible at 2191, 2207, 2327 and 2407 nm, which could correspond to alcohols or fatty acids [50,53].

#### 3.2.2. NIRS Spectrum of *Trebouxia jamesii*

Figure 8A shows the mean spectrum of the alga *Trebouxia jamesii,* and the second derivative with the wavelengths of the most relevant peaks is shown in Figure 8B. The profile is almost identical to *T. lynnae* with little peak wavelength variation. Three peaks related to free water are observed at 1157, 1409 and 1899 nm [48]. In addition, peaks attributed to H_2_O asymmetric vibration at 1343 nm and S_2_ water at 1462 nm are again clearly visible [47,49], as well as peaks related to saturated (1204 nm) and unsaturated fatty acids (2123 nm) [50], and polyols/glucides (2282 nm) [52,54]. Other peaks related with aliphatic chains are visible at 2191, 2209, 2328 and 2409 nm and could correspond to polyols or fatty acids [50,53]. However, a characteristic and distinctive band is seen around 1700 nm.

#### 3.2.3. Subtraction Spectrum of *Trebouxia lynnae* and *Trebouxia jamesii*

The subtraction spectrum was performed by subtracting the absorbance values of the mean spectrum of *Trebouxia jamesii* from the absorbances of the spectrum of *Trebouxia lynnae* (Figure 9). Areas above zero indicate that *Trebouxia lynnae* has a greater contribution of compounds that absorb at these wavelengths. The higher contribution of unsaturated fatty acids in *T. lynnae* spectra indicated by the bands around 1676 nm, and from 2100 to 2170 nm linked to the CH stretch (–CH=CH–) must be highlighted, since these bands have been used to quantify the unsaturated fatty acids in the microalga *Chlorella vulgaris* [50]. Areas below zero indicate that *Trebouxia lynnae* has a lower contribution of these compounds. Thus, *T. jamesii* shows bands due to symmetrical and asymmetrical H_2_O vibrations and water solvation shells (1336 to 1376 nm) [47] and possesses higher amounts of water (band around 1920 nm) [48]. In addition, the metabolomic fingerprint of this species shows a higher contribution of saturated fatty acids (band 1195 to 1215) [50].

### 3.3. Phycobiont Dehydration Study

#### 3.3.1. Spectra of Dehydrated *Trebouxia lynnae* and Principal Components Analysis (PCA)

As explained, lichen and lichen microalga are adapted to anhydrobiosis and tolerate extreme desiccation. The mechanisms underlying this adaptive trait are still poorly elucidated. In Figure 10, we can observe that the greatest water loss occurs in the first hour (>60%), and then the loss is slower until it reaches a minimum at 3 h (20% of fresh weight), where it stabilizes.

The mean NIR spectrum was taken for each dehydration time and Figure 11A was obtained (N = 21). As expected, the spectra are very different, especially in the water bands whose intensity decreases as desiccation progresses. In Figure 11B, the second derivative has been applied to the mean spectrum of 24 and 48 h desiccated algae (the spectrum of the fresh alga is represented in Figure 7). Both at 24 and 48 h, we observe that peaks related to free water even disappear (1151, 1409 and 1900 nm).

Another remarkable feature is that the metabolomic profile of *T. lynnae* is quite different at 24 and 48 h, and the position of some peaks shift or change in intensity. The peak at 1425 nm at 24 h, corresponding to the water hydration band, shifts to 1442 nm at 48 h, an area related to the band of S_1_ water molecules with one hydrogen bond [47]. In addition, a shift of the peak at 1685 nm at 24 h, specifically related to unsaturated fatty acids in microalgae [50], to higher wavelengths such as 1718 nm at 48 h, could indicate a decrease in the degree of fatty acid unsaturation. On the other hand, the peak observed around 2270 nm at 48 h could be related with a higher presence of alcohols or sugars [52,54], as well as the peaks in the bands from 2110 to 2400 nm [53]. At 2370 nm, a peak related to aqueous protons has also been described [56].

Figure 12A shows the PCA score plot in the space of the first two principal components (PCs), explaining the 96% of the total variance. Samples are represented using different colors according to their dehydration time; it is possible to notice a rather clear circular trend from fresh samples in the lower right quadrant (positive PC1 and negative PC2 values) to samples dehydrated for a few hours at positive PC2 values, and then samples dehydrated for 24 or 48 h that fall at negative PC1 values.

According to this observation, PC1 and PC2 seem to explain information related to the dehydration of the samples. Looking at the loading profile on PC1 in Figure 12B, it is possible to see two positive bands around 1420 nm and 1920 nm which have higher loadings on this PC. Both bands are related with water molecules, and the first one is specifically related with the water hydration band [47,48]. The highest loading belongs to a band centered at 2420 nm which could be related to aliphatic chains [53], whereas the negative loadings around the band centered at 2260 nm could be related to alcohols [52,54].

Two bands around 1415 and 1880 nm are present in the loading profile on PC2 (Figure 12C), but in this case, they are negative confirming the opposite contribution of PC1 and PC2 on the dehydration process, according to the trend observed in the score plot. However, positive loadings are dominant at bands corresponding to polyols/sugars (2110–2400 nm) [53] and aliphatic chains (2480 nm) (2225–2492 nm) [52].

#### 3.3.2. Subtraction Spectrum of *Trebouxia lynnae* Fresh and Dehydrated for 48 h

The subtraction spectrum was performed by subtracting the absorbance values of the mean spectrum of desiccated *T. lynnae* at 24 h from the absorbances of the spectrum of fresh *T. lynnae* (Figure 13A). Areas above zero indicate that fresh *T. lynnae* cells have a greater presence of compounds that absorb at these wavelengths, mainly free water (1406 and 1879 nm). Areas below zero indicate that the desiccated *T. lynnae* spectrum has a stronger contribution of compounds that absorb at these wavelengths. We observe several peaks at the higher wavelengths where bands associated to alcohols (2110–2400 nm) [53], sugars (2225–2325 nm) [52], and fatty acids (2100–2170 nm for unsaturated and 2300–2370 nm for saturated) [50] have been described.

Figure 13B shows an equivalent subtraction spectrum, but using the absorbances of the 48 h desiccated algae. At 48 h, algae also possess less free water than fresh algae, but it is clearly seen that both figures are different, indicating that metabolomic changes occur during anhydrobiosis.

Table 1 gathers the most frequent bands detected during the dehydration studies. As expected, many of them correspond to water, but also polyols/sugars and fatty acids might be relevant during desiccation for the microalga *T. lynnae*.

#### 3.3.3. Aquaphotomics

Different structural types of water bands can be elucidated around 1450 nm, where the first overtone of the O-H bond is located. To study this range in detail (1300–1700 nm), the spectra of fresh and dehydrated *T. lynnae* were analyzed and transformed with the second derivative (Figure 14).

With the aid of our previous studies (Table 1) and the literature, six peaks corresponding to different molecular species of water were identified: S_r_ (1346 nm) clustering of protonated water molecules; S_0_ (1406 nm) free water; S_1_ (1443 nm) water molecules with hydrogen bonds; S_2_ (1464 nm) water molecules with two hydrogen bonds; S_3_ (1490 nm) water molecules with three hydrogen bonds; and S_4_ (1652 nm) water molecules with four hydrogen bonds [44,45] (see Table 2).

To calculate the relative absorbance of the six water species, crude spectra of each desiccation time were corrected with EMSC before calculating the average. Then, the ratio of the sum of all absorbances at each dehydration time was calculated (Figure 15A). In this way, the individual contribution of each water species in relative weight form to the water structure can be compared. The most important species is S_0_, then, in decreasing order, S_4_, S_2_, S_1_, S_r,_ and S_3_. During dehydration, free water (S_0_) loses relative weight while other species such as S_r_, S_1_, and S_2_ gain relevance. At 4 h, there is a turning point where all water species gain relative weight, while S_4_ decreases.

Absolute absorbance of each water species was referred to the fresh state (100%) to compare the situation at each dehydration point. In the case of S_r_ (Figure 15B), the level progressively decreases up to 4 h, at 2% of the initial content. It then increases sharply to 40%, where it remains. In the case of free water (S_0_) (Figure 15C), the content progressively decreases until it reaches 5% of the initial content, stabilizing at 24 h. For both S_1_ (Figure 15D) and S_2_ (Figure 15E), we observe a progressive decrease up to 48 h. In the case of S_1_ at 48 h, it has 43%, and S_2_, 15% of its initial content. In S_3_ (Figure 15F), it decreases abruptly to 2 h, recovering slightly thereafter. Finally, S_4_ increases until 4 h (Figure 15G), where it is the major type of water. It is the only water species whose quantity increases with dehydration.

## 4. Discussion

This is the first report of the application of NIRS to elucidate the metabolomic profiles of lichen thalli together with their phycobionts. The acquisition of the NIR spectra can be completed directly in relatively small amounts of sample without pre-treatment, and in an extremely fast and convenient way. Despite *Ramalina farinacea* and *Lobarina scrobiculata* thalli being collected in an air-dry state, during a dry weather period, and kept in this physiological state at the laboratory, the major peaks of water at 1359, 1435, and 1920 nm still show a high relevance. It is interesting to note that 1359 nm has been related to water solvation shells [49] and 1435 nm to water dimers (S_1_) [47] which have been involved in the desiccation tolerance of the resurrection plant *Haberlea rhodopensis* [46]. Peaks related to lipids and polyols/glucides can also be identified. Lichen thalli are very peculiar organisms that are adapted to anhydrobiosis. They are really dry in physiological conditions with only a 2–10% of water content [58,59]. Lipid content has been reported to vary from 18 to 34 mg/g dry weight [60]. These levels may increase under warm dry weather conditions, reaching 154 mg/g fresh weight in *Evernia prunastri*, a similar species to *R. farinacea* [61].

Surprisingly, the metabolomic profile of *R. farinacea* in the dry state proves that its spectrum has a higher molecular water contribution than *L. scrobiculata*’s. Lichens with cyanobacteria as a photobiont have been reported to possess a higher water-holding capacity (WHC), measured as the difference of the wet mass (fully hydrated) and dry mass (forced desiccation at 65 °C for 72 h). Specifically, *L. scrobiculata* has been found to have a greater WHC than *R. farinacea* [62]. Thus, its NIRS spectrum could have been expected to show a higher water contribution, but this has not been the case. However, we have performed our analyses in air-dry thalli, whereas Trobajo et al. submitted thalli to a forced, non-physiological desiccation. *L. scrobiculata* was also reported to have less tolerance to desiccation than *R. farinacea* [27]. The higher contribution of S_2_ water observed in air-dry *R. farinacea* thalli has been linked with the importance of protein stabilization [46]. This fact could be related to this species’ higher tolerance to desiccation, explaining this apparent contradiction [22]. An aquaphotomics study comparing the physiological transition from fully hydrated to desiccated thalli could shed light on this biological trait of lichens. The important differences in the lipidic content and profile observed between both species may be related to energetic metabolism and deserves further investigation.

The clustering of *R. farinacea* thalli seen in the PCA analysis, according to location and orientation, proves the potential of NIR spectroscopy to detect differences among closely related populations, as seen in other studies [63]. North–south groupings may be due to microenvironmental humidity levels, as suggested by PCA loadings around 1460 (S_2_ water molecules) and 1929 nm [47,48]. The significant differences observed in thalli from distant sites within the same population, Silla de Felipe II (40°34′6″ N, 4°9′9″ W, 1057 m a.s.l.) is located at a slightly higher altitude than Ermita de la Virgen de Gracia (40°34′23″ N, 4°9′15″ W, 969 m a.s.l.), are related to S_1_ water molecules instead. Furthermore, PCA negative loadings show an inverse relationship with bands related to glucides and lipids, and specifically to a band near 1680 nm that is contributed by a CH stretch (-C=C-) and has been used to quantify unsaturated fatty acids in several oleaginous microalgal species of the genus *Chlorella* [50,54].

While the two major water peaks observed in the thalli of *R. farinacea* are related to hydrogen bonded water (S_1_), three peaks related to free water are seen in its fresh phycobionts where cells are fully hydrated (1151, 1409 and 1900 nm). In this case, all the peaks have been attributed to free water molecules in the literature [48]. Additionally, peaks attributed to the H_2_O asymmetric vibration and S_2_ water are also present [47,49].

As for other metabolites, both profiles show an important contribution of lipidic aliphatic chains (both saturated and unsaturated) and polyols/glucides. The lipidic content of the phycobiont *Asterochloris erici* Achmadjian, which belongs to a genus of microalgae closely related to *Trebouxia*, has been reported to be relatively high, reaching 10% of dry weight [64]. On the other hand, the accumulation of polyols, including sugars, has been shown to prevent the cell membrane from damage and helps form the glassy state during dehydration [65]. Moreover, the presence of lipid–sugar associations, namely polar oligogalactolipids, seems to protect the structure of the microalgal thylakoid during desiccation–rehydration cycles [66].

The recognizable similarity of *Trebouxia lynnae* and *T. jamesii* NIRS fingerprints is in accordance with two species belonging to the same genus and sharing similar habitat preferences. Nonetheless, small changes in water content and lipidic profile are evidenced using spectral subtraction, indicating that while the former has a higher content of unsaturated fatty acids, the latter accumulates more water associated to solvation shells.

During anhydrobiosis, cells are supposed to be in a state of suspended life in which metabolic processes are paralyzed [65]. However, NIR spectra demonstrate that the metabolomic profile of *T. lynnae* drastically changes from 4 to 24 h, and then again at 48 h, involving changes in water conformation, as well as lipids and polyols/glucides, that could be related with the aforementioned polar oligogalactolipids [66]. A recent study using dynamic mechanical thermal analysis and xanthophyll cycle pigment epoxidation was used to assess enzyme activity, showing that subtle differences in water content has important consequences on the metabolic reactions that occur during the desiccated state. Candotto et al. [67] have recently shown evidence that while enzyme activity ceases when the cytoplasm of lichen cells reach the glassy state, a certain degree of enzymatic activity is still possible in a “rubbery” state at slightly higher water contents [67]. Interestingly, our data seem to contribute to supporting this hypothesis.

The aquaphotomics study performed on *T. lynnae* indicates that available water changes from one molecular compartment to another as desiccation advances. During the first two hours, free water (S_0_) decreases, whereas water with four hydrogen bonds (S_4_) increase. Free water molecules enable most chemical, biological, and physiological reactions and provide turgor volume. The allocation of water to higher bonding compartments correlates with cytoplasmatic vitrification, where polyols and nonreducing sugars play a fundamental role both in microalgae and plants [46,65]. This conformation of water is consistent with described dehydration mechanisms, such as the accumulation of sugars, volume contraction, cell wall folding, or subdivision of vacuoles [68,69,70,71]. One-hydrogen- bond water molecules (S_1_) also gain relative weight in this first phase of desiccation. As explained above, this has been related with desiccation tolerance of the resurrection plant *H. rhodopensis* [46]. At 3 h, the proportion of water in protonated clusters (S_r_) decreases, almost disappearing at 4 h, which agrees with the allocation of water to high bonding conformations seen during the first two hours. On the other hand, the relative proportion of water with two hydrogen bonds (S_2_) stays stable through the whole desiccation process. Kuroki et al. [46] have related this observation in *H. rhodopensis* with the importance of protein stabilization, since this is the most abundant water species around proteins.

Similarly to what is seen in the whole NIRS spectrum, the variation in the structure of water in cells desiccated in 24 h is striking compared to 48 h. Protonated water clusters (S_r_) significantly increase, both in relative and absolute terms, probably at the expense of the highest bonding conformation (S_4_). S_3_ species also show a slight increase, while the rest of the water species remain at similar levels. At 48 h, S_4_ recovers slightly without further remarkable changes.

## 5. Conclusions

NIRS metabolomic fingerprinting is revealed as an extremely fast and versatile tool for the study of differences between lichen taxa, but also for the study of phenotypical changes induced by the microenvironment. Our data point to a higher water retention capacity in the natural dry state of *Ramalina farinacea* than *Lobarina scrobiculata*, related with S_2_ water molecules and protein stabilization, providing a molecular mechanism for its higher desiccation tolerance.

The *R. farinacea* phycobionts *Trebouxia lynnae* and *T. jamesii* show remarkably similar NIRS fingerprints, where water strongly contributes, as well as lipids and polyols/glucides. These metabolomic groups have been previously linked to desiccation tolerance and anhydrobiosis adaptation. Aquaphotomics of *T. lynnae* desiccation demonstrates that this microalga allocates water molecules to high-bonding conformations during desiccation as recently described for the resurrection plant *Haberlea rhodopensis*. Data also show that this phycobiont modulates water structure and metabolomic profile even during anhydrobiosis, supporting the existence of a cytoplasmatic “rubbery” state during vitrification that allows enzyme activity.

## Figures and Tables

**Figure 1 microorganisms-10-02444-f001:**
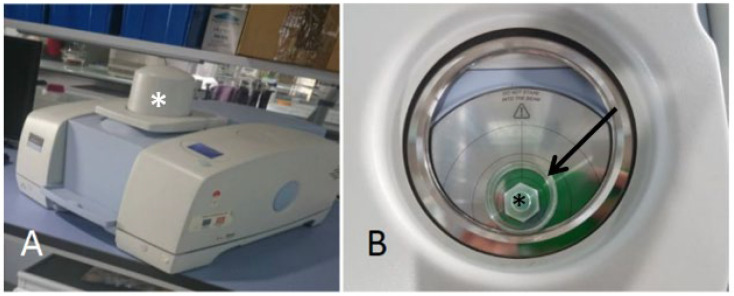
FT-NIRS Fourier transform near-infrared spectrometer. (**A**) General view. The NIRA accessory is covered by an opaque cap (marked with an asterisk) during measurements to avoid environmental interferences. (**B**) Below the cap stands the plate holder accessory.

**Figure 2 microorganisms-10-02444-f002:**
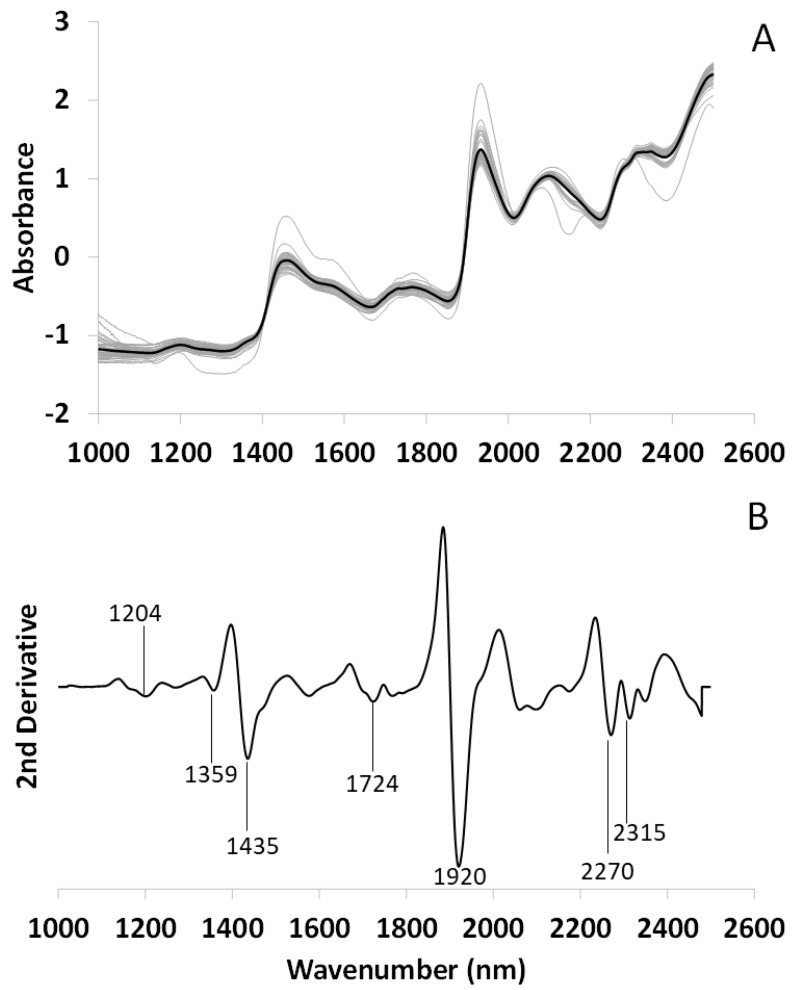
FT-NIRS spectrum of *Ramalina farinacea* thalli in a physiological air-dry state. (**A**) Mean spectrum (black line) and its replicates (N = 40, gray lines) after SNV transformation. (**B**) Second derivative of the mean spectrum, the wavelengths of the main peaks revealed have been noted for better clarity.

**Figure 3 microorganisms-10-02444-f003:**
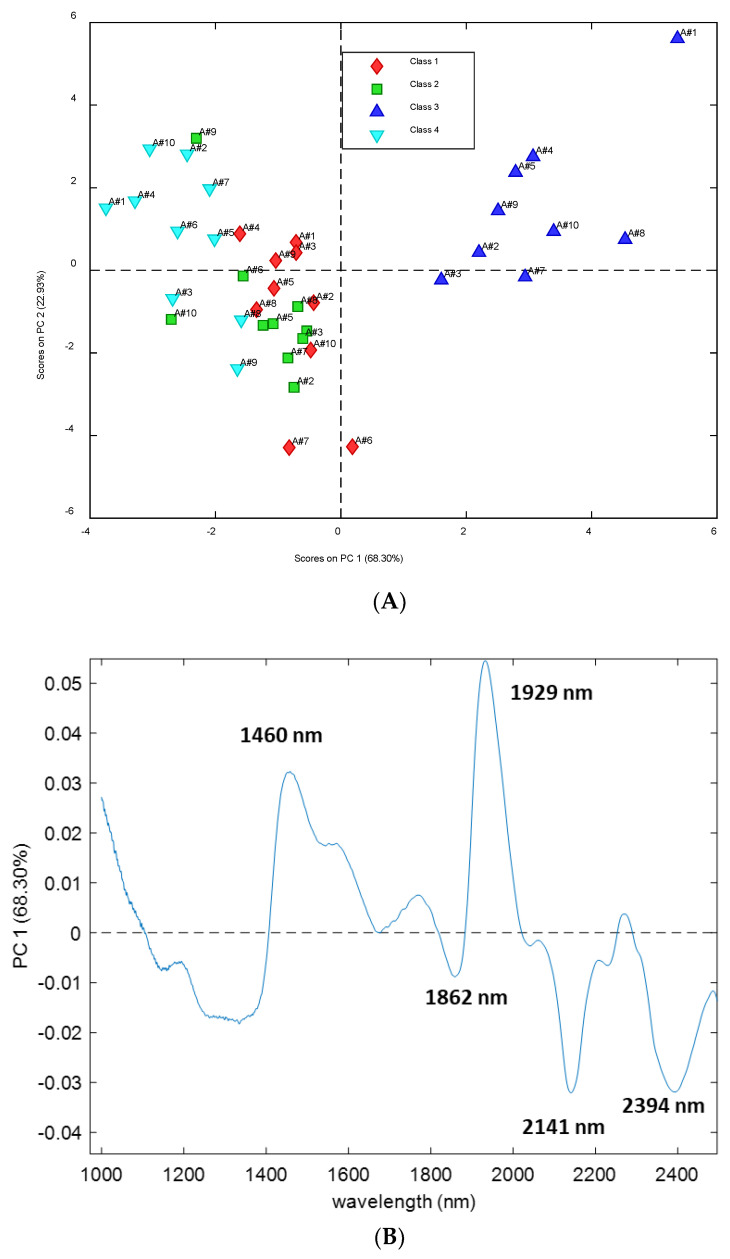
Principal component analysis of the spectra of *Ramalina farinacea* thalli. (**A**) Score plot in the space PC1-PC2 without one replica of Silla North. Samples are represented using different colors according to their site and orientation (Class 1 = Ermita North; Class 2 = Ermita South; Class 3 = Silla North; Class 4 = Silla South). (**B**) Loading profile on PC1.

**Figure 4 microorganisms-10-02444-f004:**
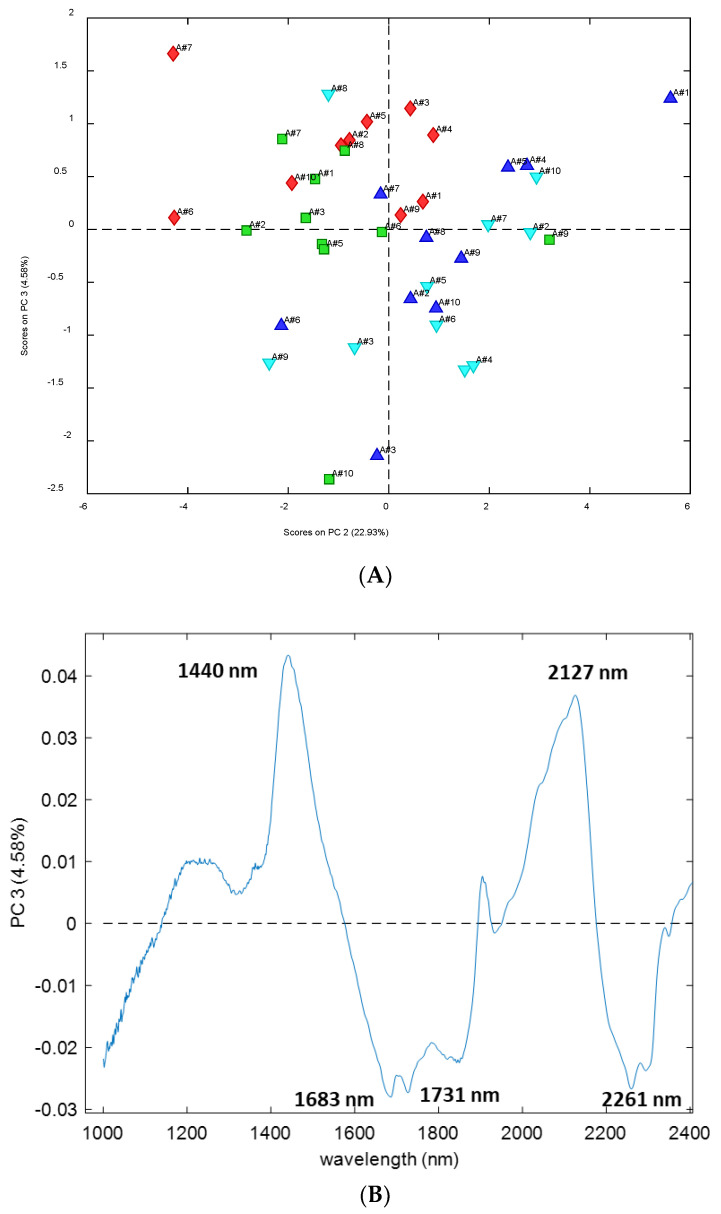
Principal component analysis of the spectra of *Ramalina farinacea* thalli. (**A**) Score plot in the space PC3-PC2. Samples are represented using different color according to their site and orientation (Class 1 = Ermita North; Class 2 = Ermita South; Class 3 = Silla North; Class 4 = Silla South). (**B**) Loading profile on PC3.

**Figure 5 microorganisms-10-02444-f005:**
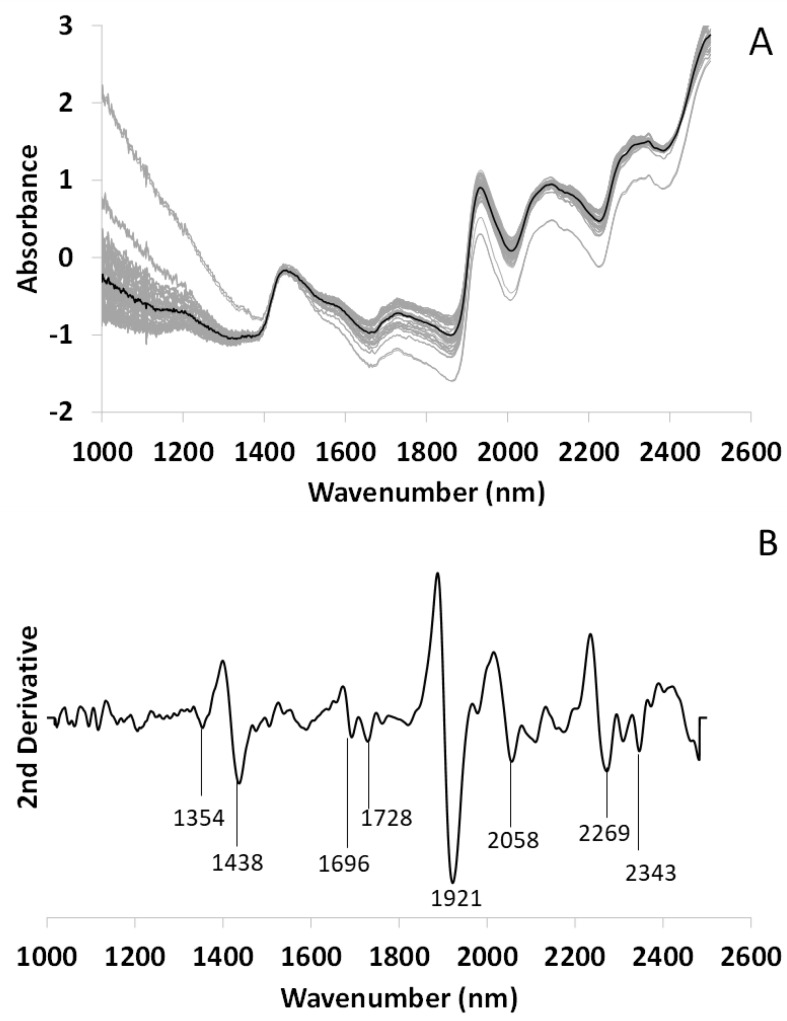
FT-NIRS spectrum of *Lobarina scrobiculata* thalli in dry state. (**A**) Mean spectrum (black line) and its replicates (N = 96, gray lines) after SNV transformation. (**B**) Second Savitzky–Golay derivative (Order 3, Points 15) of the mean spectrum with their respective metabolomic assignments.

**Figure 6 microorganisms-10-02444-f006:**
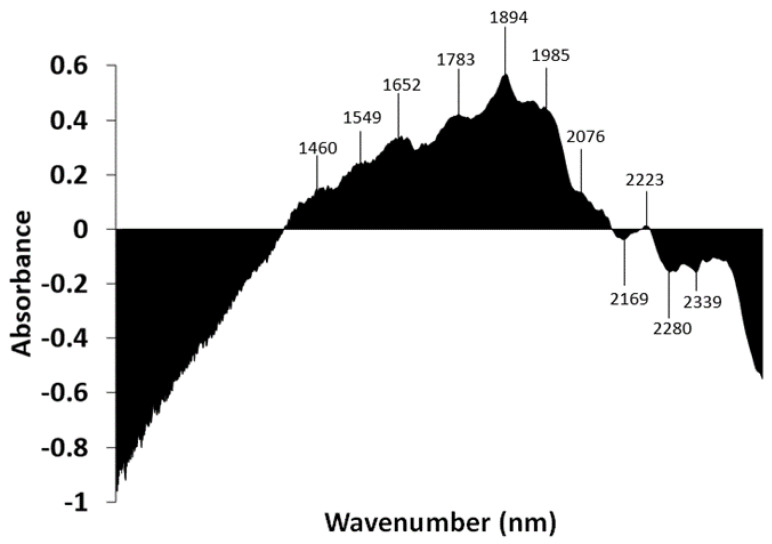
Subtraction spectrum. Spectra of each species were pretreated with SNV and then averaged. The absorbances of the mean spectrum of *Lobarina scrobiculata* were subtracted from those of *Ramalina farinacea*. The X-axis ranges from 1000 to 2500 nm.

**Figure 7 microorganisms-10-02444-f007:**
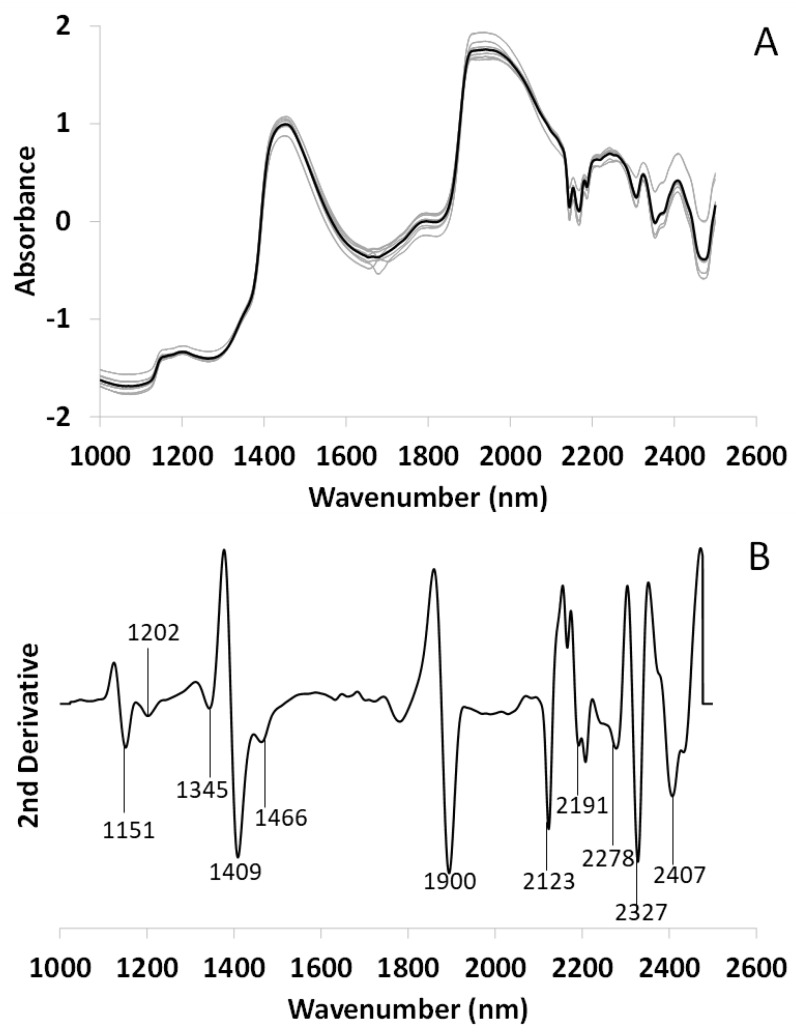
FT-NIRS spectrum of *Trebouxia lynnae* fresh alga. (**A**) Mean spectrum (black line) and its replicates (N = 21, gray lines) after SNV. (**B**) Second Savitzky-Golay derivative (Order 3, Points 10) of the mean spectrum with their respective metabolomic assignments.

**Figure 8 microorganisms-10-02444-f008:**
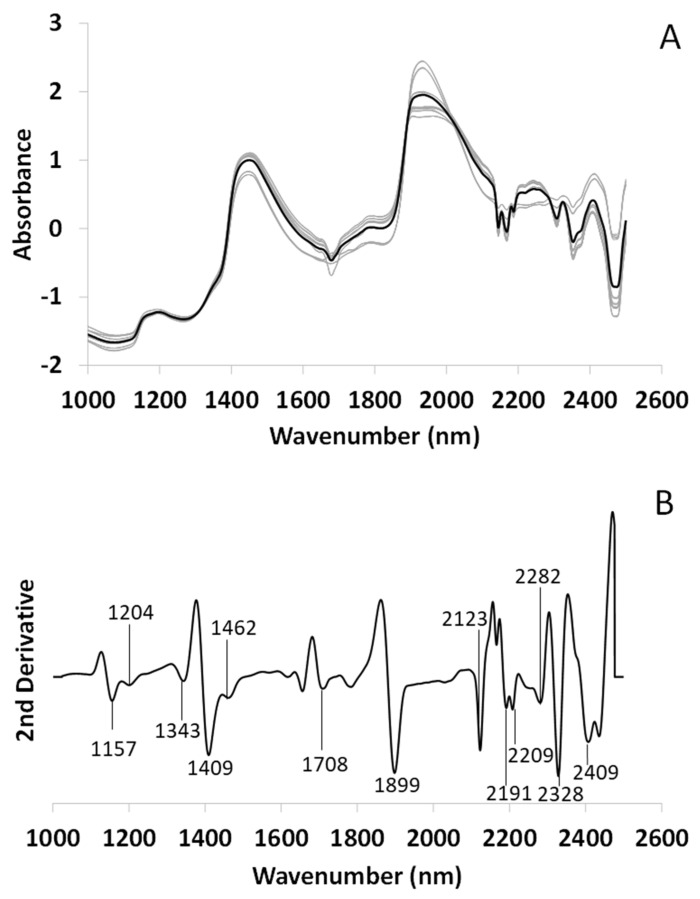
FT-NIRS spectrum of *Trebouxia jamesii* fresh alga. (**A**) Mean spectrum (black line) and its replicates (N = 21, gray lines) after SNV. (**B**) Second Savitzky–Golay derivative (Order 3, Points 10) of the mean spectrum with their respective metabolomic assignments.

**Figure 9 microorganisms-10-02444-f009:**
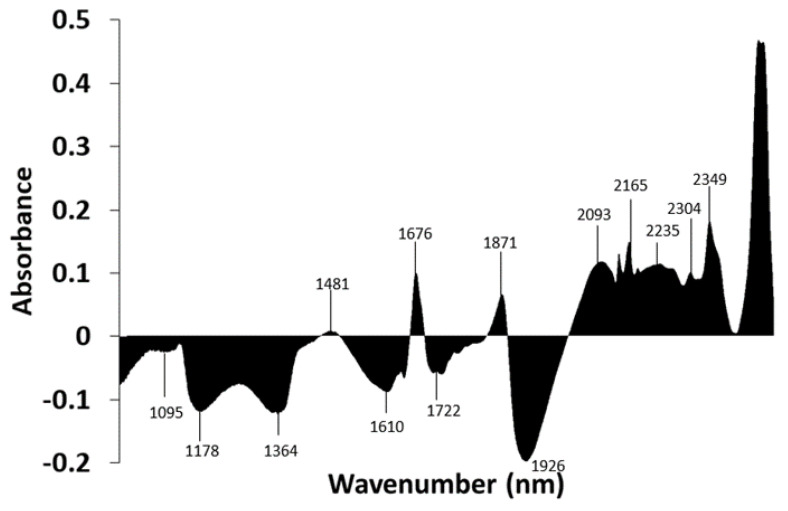
Subtraction spectrum. The absorptions of the spectrum of *Trebouxia jamesii* were subtracted from the absorptions of *Trebouxia lynnae*. The X-axis ranges from 1000 to 2600 nm.

**Figure 10 microorganisms-10-02444-f010:**
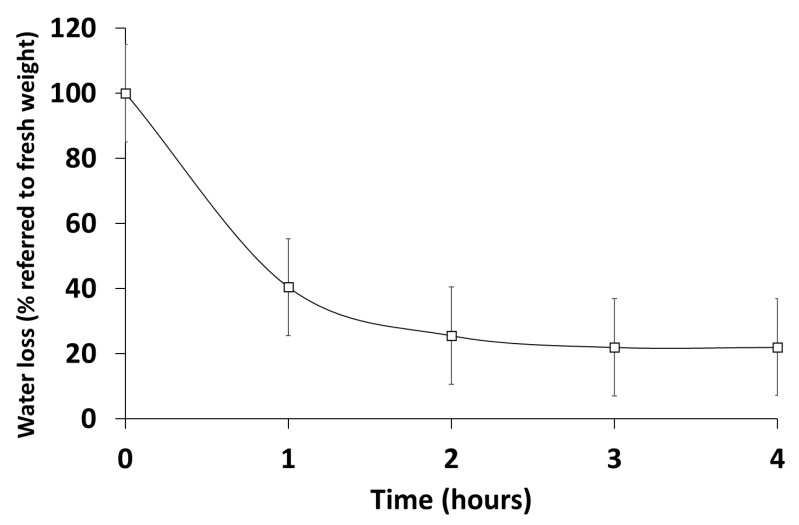
*Trebouxia lynnae* dehydration curve assessed as weight loss referred to fresh algae.

**Figure 11 microorganisms-10-02444-f011:**
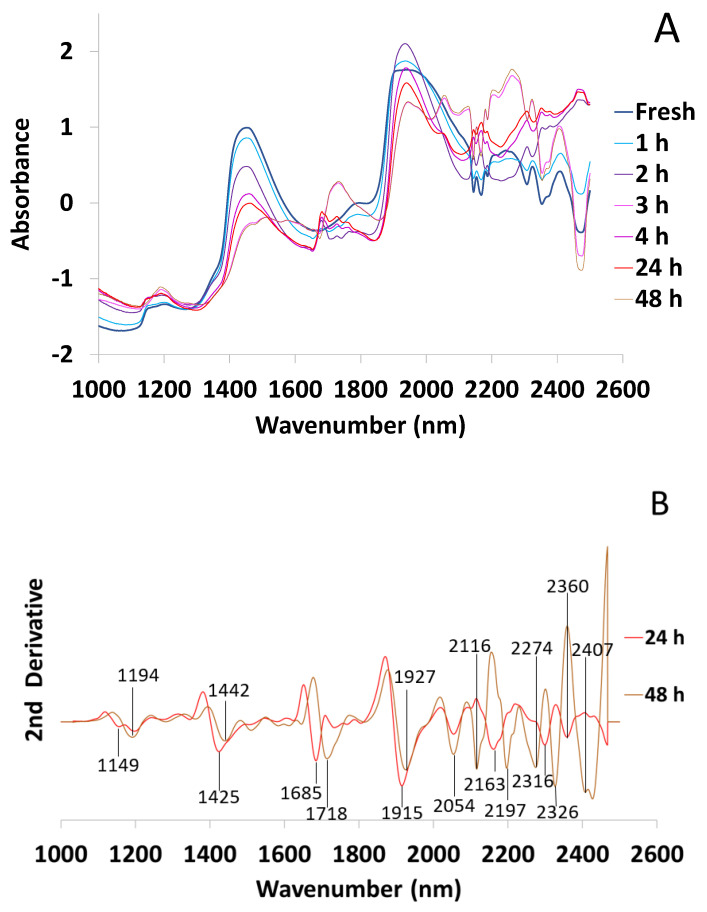
FT-NIRS spectra of *Trebouxia lynnae* at different dehydration times with EMSC (N = 21). (**A**) Mean spectra at each time point. (**B**) Second derivative of the mean spectra corresponding to 24 h and 48 h dehydration times.

**Figure 12 microorganisms-10-02444-f012:**
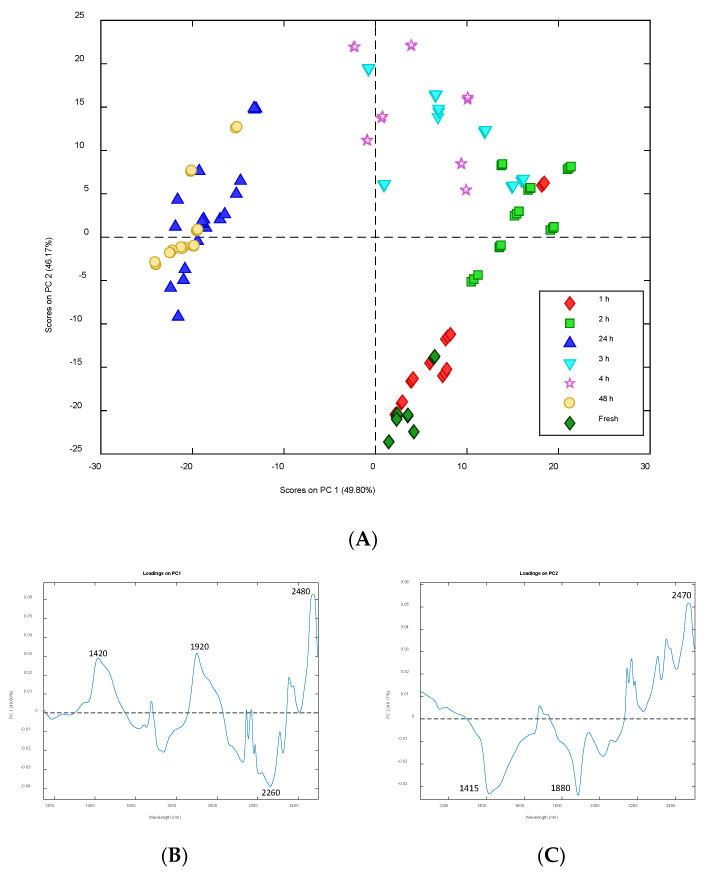
Principal component analysis of the spectra of *Trebouxia lynnae* cultures at different dehydration times. (**A**) Score plot in the space PC1-PC2. Samples are represented using different colors according to their dehydration time. (**B**) Loading profile on PC1. (**C**) Loading profile on PC2.

**Figure 13 microorganisms-10-02444-f013:**
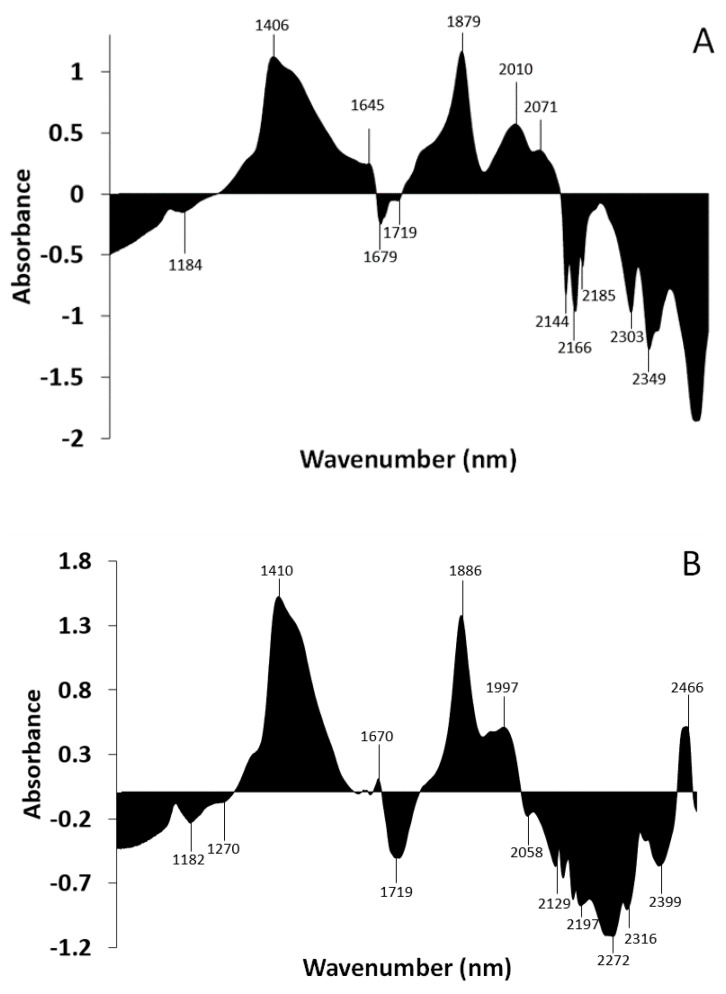
Subtraction spectrum. The absorptions of the spectrum of *Trebouxia lynnae* dehydration of 24 h from *Trebouxia lynnae* fresh (**A**) and of 48 h from *Trebouxia lynnae* fresh (**B**). The X-axis ranges from 1000 to 2500 nm.

**Figure 14 microorganisms-10-02444-f014:**
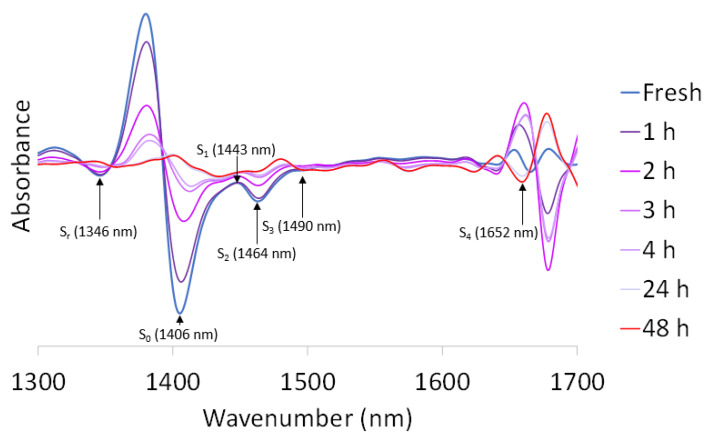
Second derivative of the mean spectrum of *Trebouxia lynnae* for the range 1300–1700 nm. The spectra were corrected with EMSC before mean calculation.

**Figure 15 microorganisms-10-02444-f015:**
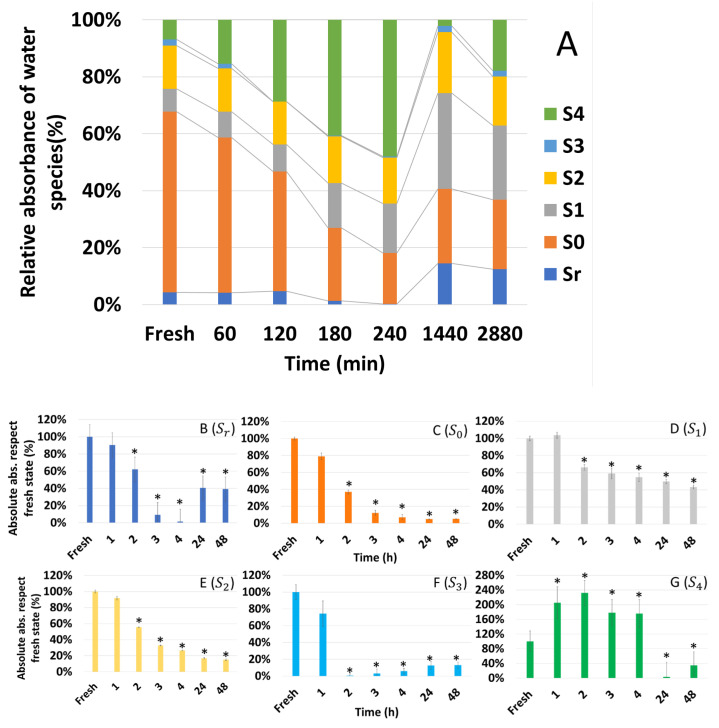
Dynamics of the different water species during the dehydration of *Trebouxia lynnae*. (**A**) Relative absorbance of the water species expressed as a percentage of the total. Graphs (**B**–**G**) show the evolution of the absolute absorbance with respect to the fresh state at time 0 (100%) of the species S_r_, S_0_, S_1_, S_2_, S_3_ and S_4,_ respectively. The asterisk indicates statistically significant differences from fresh algae (*p* < 0.05).

**Table 1 microorganisms-10-02444-t001:** Likely metabolomic groups of the most relevant bands during the dehydration of *T. lynnae*.

Band (nm)	Source	Assignment
1149–1151	2nd derivative fresh 2nd derivative 24 h	Water [48]
1202	2nd derivative fresh	Lipids [50]
1180–1194	2nd derivative 48 h Subtraction 48 h PCA	Lipids [51]
1345	2nd derivative fresh	Protonated clusters [47] Water solvation shell [49]
1406–1415	2nd derivative fresh Subtraction F-24 Subtraction F-48 PCA	Free water [49]
1420–1425	2nd derivative 24 h PCA	Water hydration band [49]
1442	2nd derivative 48 h	S_1_ water [49]
1466	2nd derivative fresh	S_2_ water [47] Hydronium involved in sugar–water interaction [49]
1685	2nd derivative 24h	Unsaturated fatty acids [50]
1718	2nd derivative 48h	Lipids [51]
1760	PCA	Lipids [50]
1879–1920	2nd derivative fresh 2nd derivative 24 h 2nd derivative 48 h Subtraction F-24 Subtraction F-48 PCA	Water [48]
2054	2nd derivative 24 h 2nd derivative 48 h	Lipids [51]
2130–2197	2nd derivative 48 h Subtraction fresh-48 h PCA	Unsaturated fatty acids [50]
2260–2272	Subtraction fresh-48 h PCA	Polyols/glucides [52,54]
2300–2350	Subtraction fresh-48 h PCA	Lipids [51]
2450–2480	Subtraction fresh-24 h Subtraction fresh-48 h PCA	Aliphatic chains [52]

**Table 2 microorganisms-10-02444-t002:** Absorbance wavelengths for the different molecular species of water with their respective reference values.

Water Type	S_r_	S_0_	S_1_	S_2_	S_3_	S_4_	References
Wavelength (nm)	1351	1410	1439	1456	1506	1642	Maeda et al. 1995 [44]
	1346	1411	1441	1462	1490	1650	Segtnan et al. 2001 [45]
	1346	1412	1441	1462	1490	1650	Kuroki et al. 2019 [46]
	1346	1403	1433	1465	1497	1650	de las Heras 2022 [57]
	1346	1406	1443	1464	1490	1652	Current study

## Data Availability

Not applicable.

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
