# Peer review of "Near-Infrared Metabolomic Fingerprinting Study of Lichen Thalli and Phycobionts in Culture: Aquaphotomics of Trebouxia lynnae Dehydration"

_microorganisms, 2022, doi:10.3390/microorganisms10122444_

Round 1

Reviewer 1 Report

Dear authors, 

I have read this manuscript with great interest, and I found that it has amazing potential. But you have to describe better the methods you used for data analysis, and I strongly advise to re-do some of the analysis. The results are presented in a very chaotic manner, and I found many of the interpretations to be questionable. Please follow my advices, especially when it comes to how to present and organize your results. You will have much clearer picture when you do it the way I suggest.  Since I follow similar topics, I see in your data huge potential and my advice is to consider and perform the major revision I am suggesting, because I am afraid that you will found yourselves in the position to work further on this data, and then, if the paper is publish as it is now, to have to actually correct and contradict yourselves in future publications. I have left many comments and suggestions for the improvements in the paper, that you can find in the notes of the pdf attached with this letter. You have a possibility to produce a groundbreaking publication, so please do not hurry with it. Take time to read more about the assignments of the bands in the NIR spectra, and especially the aquaphotomics literature. 

Best regards & good luck!

Author Response

We thank reviewer 1 for his/her appropriate comments and corrections which have helped us improve the manuscript. We are pleased to see that they are limited to specific aspects which have been corrected in the new manuscript. We also appreciate the generous and optimistic encouragement about our first incursion in the application of NIRS for the study of lichen biology.

Explanations or comments about important aspects issued by the reviewer follow below.

Abstract

The whole abstract has been revised and the reference to strongly hydrogen bonded water molecules has been better explained

Introduction

All the suggestions regarding expression or content have been accepted and conflicting phrases have been modified following the reviewer’s advice

Materials and Methods

Line. 147. I think it would be good to add the information about the conditions in laboratory if you have them by any chance (if the bench was in the laboratory, this was not very clear to me). I mean the conditions such as temperature and humidity, because I think those are relevant for the drying process

The thalli were all collected during a dry period. Therefore, they were in the physiological dry state (dehydrated). It is a usual practice among lichenologists to leave the thalli in paper bags on the laboratory bench, where the conditions are stable, up to the next day when they are either processed or frozen.

We have added the average range of temperature and humidity in our lab.

Line. 174. If by any chance you have a photograph of how you performed the measurements on thallus and how blanks were taken, I think it would be good to add in this section. I do not have a clear image in my head based on your description, and I think it would be helpful to add a photo, the journal does not limit the number of figures so why not

A photograph has been added.

Line 185. Please add references, i.e. original publications where the treatments you used were first described. Also, even though I believe it will become clear when I start to read the results, it is not clear from this description whether you performed all these treatments together, or you tried to find which is the optimal or what was the purpose. What was done after the preprocessing, did you used those preprocessed spectra to find the difference spectra, perform PCA etx.. or just to extract the peaks. This information is missing.

The whole subsection has been improved to respond to the reviewer’s questions. Also, the figure legends of each figure have been completed for clarification.

Line 253. Please add information about PCA in Data analysis section. Instead of the section about pretreatment, call it spectral analysis and describe nicely what you did.

Done as requested

Line 202. I dont think this is the best approach to chose. As can be often read in the aquaphotomics papers, people perform several types of analyses PCA, PLSR, SIMCA, difference spectra etc in order to find the CONSISTENTLY REPEATING absorbance bands in loadings of the PCA, regression vectors of PLSR, discriminating power of SIMCA, negative peaks in 2nd derivative etc. The bands that were found repeatedly in all these analyses were then considered important. I think with the approach you did you took the shortcut and you might have missed the absorbance bands that were really important for your case. Please reconsider to do the analysis this way also, you will have a more robust study. With what you did, I just have a feeling, we did this because Kuroki et al, did it like that and used these bands, but those bands might be SPECIFIC for Haberlea rhodopensis, you see. You might have other and more bands important for you lichens. Please consider this, since you have a wonderful dataset and you might be on the verge on some wonderful discovery. Of course, you do not have to do it for this paper, it might end up being too long, since you are also reporting metabolomic data, but consider to do it in future.

The advice of the reviewer has been followed and a deep revision of the results, including new PCA analyses has been performed.

Results

Line 253. It is not clear to me is this PCA done on the pre-processed data, and how preprocessed. Also, I think it is a bad idea to do PCA on average spectra. Show your data, show clusters of points. There will probably be some overlap, but the tendency of grouping will show

Show also loadings of the PCA, 1st and 2nd to show what are the differences between these groups. You will probably have some peaks revealed here. What is the meaning of the numbers next to the dots in the legend.

We have revised all the PCAs with the help of a new co-author, Dr. Monica Casale. All data sets were preprocessed with SNV and column centering. Now loadings are presented.

Line 256. This is absolutely unclear. Which analysis showed this? Which study?

This has been eliminated since it corresponded to preliminary work that served to establish the methods used. All the thalli were measured in the same conditions.

Line 269. Please provide the wavelengths at the peaks also, not only assignments . Maybe the assignments are wrong.

This has been revised following the reviewer’s advice.

Line 279. I wonder if these are indeed lipids? Do you have information about the average lipid content of these species. From the spectra one could think this has more lipids than water, which is pecular for me for one living system. Are you sure these were lipids? Could it be due to sugars? I know that many plants during dehydration accumulate sugars and create specific glassy state of water due to this.

Lichen thalli are very peculiar organisms and are adapted to anhydrobiosis. They are really dry in physiological conditions with only a 5-10% of water content (Lange et al, 1993) or even 2% (Kappen, 1973). Lipid content has been reported to vary from 18 to 34 mg/g dry weight (Dembitsky et al, 1992). These levels increase under warm dry weather, reaching 154 mg/g fresh weight in E. prunastri, a similar species to R. farinacea (Dertien et al. 1977). Therefore, yes, dry thalli could have a higher content of lipids than water.

However, polyols, including sugars are also very important during dehydration for both lichens and phycobionts

Dembitsky, V. M., Rezanka, T., & Bychek, I. A. (1992). Lipid composition of some lichens. Phytochemistry, 31(5), 1617–1620. https://doi.org/10.1016/0031-9422(92)83117-H

DERTIEN, B. K., de KOK, L. J., & KUIPER, P. J. C. (1977). Lipid and Fatty Acid Composition of Tree-Growing and Terrestrial Lichens. Physiologia Plantarum, 40(3), 175–180. https://doi.org/10.1111/J.1399-3054.1977.TB04053.X

KAPPEN, L. (1973). RESPONSE TO EXTREME ENVIRONMENTS. The Lichens, 311–380. https://doi.org/10.1016/B978-0-12-044950-7.50015-5

Lange, O.L., Büdel, B., Heber, U., Meyer, A., Zellner, H., and Green, T.G.A. 1993. Temperate rainforest lichens in New Zealand: high thallus water content can severely limit photosynthetic CO2 exchange. Oecologia 95: 303-313

Line 337. Maybe they are not related to aliphatic chains. For example see the following assignments.

2266.0 probably related to C-H combinations and O-H stretch overtone, reported to be associated with sugars in fruit juices Cozzolino, D., et al. "Combining near infrared spectroscopy and multivariate analysis as a tool to differentiate different strains of Saccharomyces cerevisiae: a metabolomic study." Yeast 23.14‐15 (2006): 1089-1096. 

2358.5 H2O deionized, 1st overt. Workman 2001: Handbook of Organic Compounds 

2369.7 aqueous proton [H+·(H2O)6] - Proton oscillation, 3rd overt. Headrick et al. (Mark Johnson) 2005: Science, 308: 1765. 

2684.56 Free OH in a free molecule V3 Luck, Werner AP, and Walter Ditter. "Approximate methods for determining the structure of water and deuterated after using near-infrared spectroscopy." The Journal of Physical Chemistry 74.21 (1970): 3687-3695. 

2702.7 (OH-(H2O)5) Science28 - Robertson, William H., et al. "Spectroscopic determination of the OH− solvation shell in the OH−·(H2O) n clusters." Science 299.5611 (2003): 1367-1372. 

2702.7 Free OH in a molecule whose second OH is H-bonded V3 Luck, Werner AP, and Walter Ditter. "Approximate methods for determining the structure of water and deuterated after using near-infrared spectroscopy." The Journal of Physical Chemistry 74.21 (1970): 3687-3695. 

Maybe these are all sugars and water??

The metabolomic assignments have been thoroughly revised using literature and the recommendations of the reviewer have been taken into account

Line 360. You have to say what pretreatment was performed, before these calculations. I do not see here clear patterns, and it may be because there is a difference in baseline owing to the changes in physical structure during drying. It would also be good, to actually start the Results section with showing the RAW spectra. So, then, the reader can conclude what is happening with the baseline and if the preprocessing should be performed.

Adequate PCA analyses that provide information about these items have been included to solve the doubts issued. If requested we could also provide the raw spectra as supplementary data.

Line 369. Please re-do this figure. If A) is the most important, place A above and B, C, D... below. Also add some little titles above graphs b, c, d, and so on... so that it is immediately clear what is represented on them

The figure has been organized as demanded.

Line 383. You can check this, by performing simple correlation analysis. See if the absorbance at those two bands shows negative correlation

We have eliminated this statement to avoid speculation.

Line 395. Again, here add the loadings!! Lets see what are the features of PC2 loading. It will reveal which bands are describing the process.

Loadings have been included

Line 403. Please consider the possibilities that these bands could also be assigned to water and sugars, not lipids

We have performed a careful review of the literature on NIRS regarding sugars and lipids of microalgae and fungi to contrast our initial assignments. To our understanding, the contribution of lipids seems confirmed. We have improved the results description and discussion including the appropriate explanations and reference.

Discussion

I would strongly suggest to put the results in better order. First, whatever you did, first show the raw spectra. Then difference spectra, label the peaks you see, then 2nd derivative, label the negative peaks, then PCA, provide loadings and lable the peaks you see. Then only go over ALL your results, and put the peaks you have seen during the entire analysis in a table. When you see the same peaks repeating, those peaks are important to describe what is happening. Then look at the assignments, and perform the aquaphotomics analysis, on the peaks that most often appeared. That will show you the real picture. You have wonderful data, but the entire results section is in chaos, and I can not see "a clear picture emerging" from it.

The recommended order has been implemented.

Also, when you do assignements, take a good care to google the peaks you find, for example, immediately at the beginning of the results, you had bands 1435 nm. This band is very nicely described in Malegori, Muncan et al. together with many other water bands in some study about the preservation of rice germ. It was assigned to hydronium, and implicated in sugar-water interaction.  I believe this band is very important for your samples. You also have some band around 1720 nm, and look at this:

1724.1 H3O+, 1st overt. Rhine et al. 1974: The Journal of Physical Chemistry, 78: 1406. M. Falk and P. A. Gigue` re, Can. J. Chem., 1957, 35, 1195 

1724.0 H3O+ stretching band M. Falk and P. A. Gigue` re, Can. J. Chem., 1957, 35, 1195 

This is the same information appearing in your spectra! Take care also about other assignments!

 From that paper too, you will se that band 1362 (or something similar 1364, 1360...) is hydration shells water.  There are other bands in that paper that can be of use to you. The band you usually assign to phenols, or polyols, I can not remember now, around  1460 nm, it might also be just WATER with 2 hydrogen bonds. 1354 nm I think is related to hydrated protons. Just like 1345 nm, which is definitely something with protons, look at the paper by Muncan et al. about fatty acids. Basically all those bands related to protons mean that you have a water in gaseous phase in your samples.

You repeatedly have bands around 1435,1438, and 1897, 1920 something like that as important. Focus on those! 1652 nm I believe was a mistake in Kuroki work. I dont think it is S4. 

We thank the reviewer for such detailed, pertinent and helpful comments. We have profoundly revised water related assignments as suggested and new perspectives have emerged, that we hope will meet the reviewer’s expectations.

Line. 413. Unless you are mistaken about what you think you see. Maybe you see sugars and water.

Metabolomic assignments have been thoroughly revised and supporting literature provided

Line 419. Focus on these two, and you will be amazed what you will see.

Line 424. If you provide loadings, you will see exactly what the differences are.

Loadings have been provided

Line 439. Your aquaphotomics study was weak, and did not show anything conclusive. Re-do it, please.

The whole analysis has been thoroughly revised

Line 433. If you separate the spectra into 3 spectral regions, 1000-1300, 1300-1800, and 1800-2500 and analyze them separately you will end up with more prominent bands. Each region has its specifics, and requires different spectral pretreatment. It also shows different things. Just separate those regions, and details will pop-up, trust me.

We appreciate this piece of advice. Since this work is already quite large, we will save this interesting analysis for a future work.

Line 446. I believe this is a mistake.

Revised

Line 448. 1435 is your proof for that, just go over the literature a bit

Revised

Line 451. the band 1460 (or closely related) is also a protein-water interaction band, so you can also relate it to the proteins

Revised

Line 464. You do not have any reference data on metabolites, and your assignments are tentative

Revised

Line 467. I am certain this is not a paradox. You assigned something wrongly

We now include an improved presentation of these data, together with supporting literature about the existence of a “rubbery” state during lichen vitrification at slightly higher water contents than the glassy state (Candotto et al 2021). In this state enzymatic activity would still be possible to a certain extent and might explain the changes in the metabolome observed from 24 to 48 h during anhydrobiosis. We believe this is one of the most relevant results of our aquaphotomics study.

Candotto Carniel, F.; Fernandez-Marín, B.; Arc, E.; Craighero, T.; Laza, J.M.; Incerti, G.; Tretiach, M.; Kranner, I. How Dry Is Dry? Molecular Mobility in Relation to Thallus Water Content in a Lichen. J. Exp. Bot. 2021, 72, 1576–1588., doi:https://doi.org/10.1093/JXB/ERAA521.

Line 482. S2

Line 483. Kuroki, not Sakudo

Revised

Reviewer 2 Report

This Manuscript concerns the use of Near Infrared Spectroscopy to perform the characterization of the metabolomic fingerprint of some lichens

The study is well planned and well described both in the materials and methods part and in the discussion and conclusions parts.

This paper can be accepted for publication after minor revisions and some clarifications.

The spectra were recorded in transflectance mode placing a reflector on top of the thallus: what material is the reflector made of?

I do not believe that the authors used only the table of Xiaobo et al. 2010 to identify the absorption peaks related to the metabolomic groups, because it is too general: it is possible to add references when specifics peaks are related to specific groups such as saturated and unsaturated aliphatic chains or polyols?

When the subtraction spectrum was performed by subtracting the absorbance values of the mean spectrum of L. scrobiculata from the absorbances of the spectrum of R. farinacea, were the spectra pretreated with SNV?

Reference # 53 has no corresponding citation in the text. Please check

Author Response

We are pleased to know that this Reviewer found our manuscript to be well planned and well described. The reviewer’s comments will be reproduced below together with our response.

The spectra were recorded in transflectance mode placing a reflector on top of the thallus: what material is the reflector made of?

We used the ICRA Liquid Reflector comes standard in the NIRA Transflectance Liquids Sampling Accessory (https://www.perkinelmer.com/es/product/icra-liquids-reflector-pkg-100-l1180503) just for practical reasons, to keep the thallus compacted towards the detector. According to the information provided by the manufacturer reflectors are made of an aluminium alloy and have a sand bath on the reflective side to make it a diffuse reflective surface.

I do not believe that the authors used only the table of Xiaobo et al. 2010 to identify the absorption peaks related to the metabolomic groups, because it is too general: it is possible to add references when specifics peaks are related to specific groups such as saturated and unsaturated aliphatic chains or polyols?

Since very little literature exists on NIRS metabolomics applied to lichens or phycobionts, we did the general metabolomic assignments using theoretical tables. However, we have comprehensively revised the work and added references of the most related organisms for the relevant peaks.

When the subtraction spectrum was performed by subtracting the absorbance values of the mean spectrum of L. scrobiculata from the absorbances of the spectrum of R. farinacea, were the spectra pretreated with SNV?

Yes, SNV was applied to spectra before averaging and subtracting. Material and method section has been thoroughly to clarify this aspect (see Data Analysis subsection). Figure legends have also been completed.

Reference # 53 has no corresponding citation in the text. Pleasecheck

The reference and the reference list has been thoroughly revised

Round 2

Reviewer 1 Report

Dear authors, 

The revised manuscript is excellent. I would only ask to please in the abstract provide what S1 and S2 water is. Just give full expressions, otherwise people will not know what you are talking about when they read abstract. 

Best regards

Author Response

We really appreciate the reviewer's help. We have clarified the abstract and corrected some minor wirting errors and typos along the draft.